# Preparation and Photovoltaic Performance of a Composite TiO_2_ Nanotube Array/Polyaniline UV Photodetector

**DOI:** 10.3390/polym15224400

**Published:** 2023-11-14

**Authors:** Chunlian Liu, Peipei Huang, Wei Wang, Miao Tan, Fangbao Fu, Yunhui Feng

**Affiliations:** 1Guangzhou Key Laboratory of Sensing Materials & Devices, Center for Interdisciplinary Health Management Studies, School of Chemistry and Chemical Engineering, Guangzhou University, Guangzhou 510006, China; 2112205032@e.gzhu.edu.cn (C.L.); 2005200035@e.gzhu.edu.cn (P.H.); 2School of Chemical Engineering and Light Industry, Guangdong University of Technology, Guangzhou 510006, China; gracetan0705@163.com (M.T.); fufangbao@gdut.edu.cn (F.F.)

**Keywords:** polyaniline, titanium dioxide, ultraviolet photodetector

## Abstract

TiO_2_ has great potential for application in UV photodetectors due to its excellent photoelectric response. In this work, composite nanomaterials of TiO_2_ nanotube arrays (TiO_2_ NTAs) and polyaniline (PANI) were successfully prepared on titanium sheets using an anodic oxidation electrochemical method. The results showed that the TiO_2_ NTA/PANI composite materials had excellent UV photosensitivity and responsiveness and good stability and reproducibility. This was mainly attributed to the p–n heterostructure formed inside the TiO_2_ NTA/PANI composites that hindered the recombination of photogenerated electron–hole pairs and improved their utilization of UV light. This work provides a theoretical basis for the application of metal oxides in UV photodetectors, which is important for the development of UV photodetectors.

## 1. Introduction

UV photodetectors have emerged as an increasingly popular topic in science research in recent years [1,2]. Due to their excellent performance with a high sensitivity, wide linear range, and good stability, UV photodetectors in the mid-ultraviolet and near-ultraviolet wavelength bands are widely used in important fields, including military and defense [3], biomedicine [4,5], environmental monitoring [6], and light wave communication [7]. The focus of study has steadily shifted to the utilization of UV radiation from fires or sunlight for ozone hole and fire monitoring. UV photodetectors have notable advantages over conventional infrared and other types of photodetectors, including an outstanding photosensitivity and accuracy, good concealment, a high military value, and quick presentation of detection data [8,9,10]. The preparation of wide-band semiconductor arrays, which is complex and expensive, the material’s poor stability, and the devices’ inadequate performance are just a few of the issues that remain unresolved despite the extensive research on UV photodetectors [11,12,13]. High-performance, high-stability, high-reliability UV photodetectors, particularly UV detectors that can operate in extreme and harsh conditions, are extremely important to meet the needs of various fields, and the accuracy and reliability of UV photodetectors have a significant influence on practical applications. Due to their benefits, several semiconductor materials with various bandwidths are currently employed as substrate materials for UV photodetectors [14,15,16,17,18,19,20]. TiO_2_ is one of today’s most popular wide-bandwidth semiconductors due to its benefits of a strong chemical stability, low toxicity, and environmental friendliness [21,22,23]. Rutile TiO_2_ and anatase TiO_2_ have bandwidths of around 3.0 and 3.2 eV, respectively [24]. The heterojunction formed by combining the two crystal phases can reduce the forbidden bandwidth and produce a mixed crystal effect [25], which can somewhat facilitate the separation of photogenerated electron–hole pairs. This is because the two crystal phases have different valence band energies. In ultraviolet photodetection, the aforementioned good optoelectronic features have received a lot of attention [26,27]. Among them, TiO_2_ NTAs have a highly ordered structure that can offer numerous charge transport channels, a greater specific surface area, and more UV light interaction sites, opening up a wide range of possibilities for the development of UV photodetectors with a higher performance and efficiency [28,29,30,31,32]. TiO_2_, on the other hand, has a very high internal charge complex rate as an n-type metal oxide semiconductor, and TiO_2_-based photodetectors need an external voltage to provide potential difference excitation for proper operation, which results in large device sizes and severely restricts their application in real life [33]. TiO_2_TiO_2_TiO_2_TiO_2_TiO_2_TiO_2_TiO_2_P-type conductive polymers like polyaniline (PANI) have a high electrical conductivity and good electrochemical characteristics. They absorb light well, move carriers quickly, have a high capacitance, are easy to fabricate, and are environmentally friendly [34,35,36,37]. Photogenerated carriers can be prevented from combining in p–n heterojunctions because their internal electric fields cause photoelectrons and holes to flow in different directions [38,39]. Mixing the n-type metal oxide semiconductor TiO_2_ with the p-type conductive polymer PANI creates a p–n hetero-junction [40,41,42]. The heterojunction’s automatically produced potential difference can push photogenerated electron–hole pairs apart and induce a photocurrent, giving UV photodetectors [43,44,45] a high UV sensitivity.

The titanium substrate, which has a large specific surface area, is highly stable, and can be easily recycled, is fixed with neatly organized TiO_2_ nanotube arrays (NTAs). However, TiO_2_’s large band gap, UV light responsiveness, and high electron–hole pair recombination rate require modification via doping to improve the performance. Numerous researchers have coupled TiO_2_ NTAs with the conductive polymer PANI to develop high-performance compounds. Wang et al. [46] produced a TiO_2_ nanotube array photoelectrode (PANI/TiO_2_) wrapped in polyaniline (PANI) nanosheets using anodizing and vacuum-assisted impregnation. Polyaniline nanosheets that are securely wrapped in TiO_2_ nanotubes exhibit superior light responsiveness and charge separation efficiency compared to pure TiO_2_. PANI/TiO_2_-0.1 has 0.73 times the photocurrent density of TiO_2_. PEC p-phenol degrades PANI/TiO_2_-0.1 27.7% better than TiO_2_. It was found that the polyaniline/TiO_2_ photoelectrode synergized photocatalysis and electrocatalysis. Chen et al. [47] produced a PANI/H-TiO_2_ nanotube array composite electrode utilizing constant voltage electrochemical deposition. Before making the PANI/H-TiO_2_ nanotube array composite electrode, the researchers anodized a titanium sheet. They then calcined and hydrogenated the sample in hydrogen, put it in an acetone solution containing sulfuric acid and aniline, and deposited polyaniline on the titanium dioxide nanotube array at a continuous voltage. PANI/H-TiO_2_ nanotubes exhibit excellent electron transport and charge transfer, as shown by their solution resistance of 0.1554 and charge transfer resistance of 2.723 V cm^2^. Xie et al. [48] used electrochemical potentiostatic potential polymerization to construct a PANI-TiO_2_ nanotube array composite electrode. The anodized titanium dioxide nanotube array was immersed in a hydrochloric acid solution with aniline monomer, agitated until completely dissolved, and then polyaniline was added using the potentiostat method (0.9 V) to the same solution to generate a composite electrode. They observed arrays of disordered polyaniline nanowires around TiO_2_ nanotubes, which possessed a unique microstructure, a high specific surface area, a fast ion transport channel, and long-term cycle stability. An excellent electrochemical performance was exhibited by the composite. After recombining the TiO_2_ nanotube array and PANI, the material’s specific surface area, stability, and charge separation efficiency improved. Future photoelectric detection materials could benefit from this substance. 

This study examines photodetectors and suggests combining TiO_2_ nanotube arrays with a p-type conjugated conductive polymer PANI using a two-step electrochemical method to increase the material’s spectral response range and ultraviolet light absorption. This analysis relies on previous research. A good p–n heterojunction interface inside the device improves the ultraviolet photodetectors’ response time, sensitivity, and stability, accelerates photogenerated electron–hole pair separation, suppresses carrier recombination, and boosts quantum efficiency.

## 2. Experimental Section

### 2.1. Chemicals

Aniline (C_6_H_7_N, AR) was purchased from Alfa Aesar (Shanghai, China). Anhydrous ethanol (C_2_H_5_OH, AR) and hydrofluoric acid (HF, AR) were purchased from Tianjin Damao Chemical Reagent Co., Ltd. (Tianjin, China). Concentrated sulfuric acid (H_2_SO_4_, AR) and acetone (CH_3_COCH, AR) were purchased from Guangzhou Chemical Reagent Factory (Guangzhou, China). Titanium flakes were purchased from Shaanxi Baoji Erlitai Products Co., Ltd. (Baoji, China).

### 2.2. Preparation of TiO_2_ NTAs

The titanium sheet was polished with sandpaper until the surface had a metallic luster, and then it was evenly cut to a thickness of 0.5 mm, a length of 20 mm, and a width of 10 mm. Then, the sheet was placed in acetone, anhydrous ethanol, and deionized water for ultrasonic cleaning for 30 min each, and then it was removed and dried for spares. A two-electrode system was adopted, where the pre-treated titanium sheet was placed on the electrode clamp as the anode, the platinum sheet was used as the cathode, a DC regulated power supply was used as the external power supply, and a hydrofluoric acid solution with a mass fraction of 1.5% was used as the electrolyte. After connecting the parts of the device, the samples were anodized at 20 V for 8 min, removed and rinsed in deionized water until the electrolyte was removed from the surface, and then dried. Finally, the samples were put into a muffle furnace with a heating rate of 5 °C/min and annealed at 600 °C for 3 h to produce the desired TiO_2_ NTAs.

### 2.3. Preparation of TiO_2_ NTAs/PANI Composite Nanomaterials

TiO_2_ NTAs were composited with PANI using an electrochemical method at room temperature. A three-electrode system was used at room temperature, and an aniline sulfuric acid solution with a mass fraction of 0.8% was used as the electrolyte using a 1 mol/L dilute sulfuric acid solution. The TiO_2_ NTA samples prepared above were placed on the electrode clamp as the anode, a platinum sheet electrode was used as the cathode, and a saturated mercuric glycol solution was used as the reference electrode. The electrodes were immersed in electrolyte and reacted for 1 h at a constant potential of 0.9 V using a potentiostat in an electrochemical workstation (CHI660D), with a sample interval of 0.1 s. The samples were taken out at the end of the reaction, and the surfaces of the samples were rinsed clean of the electrolyte with deionized water and dried to obtain the TiO_2_ NTA/PANI composite nanomaterials. The preparation flow diagram of the TiO_2_ NTA/PANI device is shown in Figure 1a,b.

### 2.4. Characterization

The micromorphology of the samples was evaluated using scanning electron microscopy (SEM, JSM-7001F). The crystal structure of the samples was investigated by X-ray diffraction (XRD, PW3040/60) and the molecular structure was investigated using a Raman spectrometer (LabRAM Aramis, Villeneuve d’Ascq, France).

### 2.5. Photoelectric Measurements

The photoelectric properties were tested using a UV-visible spectrophotometer (UV 2450, Shimadzu, Kyoto, Japan), a handheld dual-wavelength UV lamp (ENF260C, Spectronics, Underwood, QLD, Australia), a fluorescence spectrometer (FLS1000, Edinburgh, UK), and an electrochemical workstation (CHI660D, Shanghai, China).

## 3. Results and Discussion

### 3.1. Regulation of Process Parameters

The titanium sheet served as the anode, the aluminum sheet served as the cathode, and the electrolyte was a solution of hydrofluoric acid with a mass fraction of 1.5%. The reaction was conducted at room temperature for 8 min using the anodic oxidation method. Figure 2a–c displays the SEM images of the products obtained at oxidizing voltages of 15 V, 20 V, and 25 V, respectively. TiO_2_ NTAs were not generated on the surface of the titanium sheet when the oxidation voltage was as low as 15 V, but rather a laminated blade-like structure was observed. When the oxidation voltage was 20 V, TiO_2_ NTAs with smooth surfaces, large tube diameters, homogeneous morphologies, and stability were effectively manufactured. The TiO_2_ NTAs shattered and their surfaces became granular when the oxidation voltage reached 25 V, demonstrating that the voltage was too high at that point. A voltage of 20 V was therefore chosen as the subsequent reaction voltage. Other reaction conditions remain unchanged. SEM images of the samples obtained by oxidizing titanium sheets at room temperature for different durations are shown in Figure 2d–f, representing oxidation times of 5 min, 8 min, and 10 min, respectively. When the reaction was carried out for 5 min, it can be clearly observed that a large and uniform pore-like structure appears on the surface of the titanium sheet, and this phenomenon was considered to be a precursor to the nanotube array structure, even though it did not form a complete TiO_2_ NTA structure. When the reaction was carried out for 8 min, it was observed that complete and uniform TiO_2_ NTAs were obtained on the surface of the titanium sheet, with a uniform diameter and smooth mouth, which were the target products. With an increase in the reaction time to 10 min, the tube diameter was slightly reduced, and there was a local rupture of the tube mouth. Therefore, 8 min was chosen as the reaction time in subsequent experiments.

Figure 2g–i corresponds to electrolytes at hydrofluoric acid mass fractions of 1%, 1.5%, and 2%, respectively. When the electrolyte was hydrofluoric acid at 1% concentration, TiO_2_ NTAs were observed, but their walls were granular and slightly thinner than those of the target product. When the electrolyte was hydrofluoric acid at a concentration of 1.5%, complete TiO_2_ NTAs could be observed, and the tube wall was smoother, with a moderate thickness which was uniform and stable. When the electrolyte was 2% hydrofluoric acid, the tube wall was further thickened, and it can be observed that its thickness affected the size of the tube aperture, with the possibility of affecting the surface of the TiO_2_ NTs when in full contact with UV light. Therefore, during the preparation of TiO_2_ NTAs by anodic oxidation, the concentration of the electrolyte can directly affect the wall thickness of the resulting TiO_2_ NTs, tending to thicken as the concentration of the electrolyte increases. Considering the aim of increasing the photoelectric conversion efficiency of the UV photodetector by ensuring the material is in contact with UV light as much as possible, a hydrofluoric acid solution with a mass fraction of 1.5% was selected as the electrolyte.

### 3.2. Characterization of TiO_2_ NTA/PANI Nanocomposites

Figure 3a,c shows the top view of the TiO_2_ NTAs, which shows that the TiO_2_ NTAs were perpendicular to the substrate of the titanium sheet, and each one of them was separate and independent with a relatively uniform size. The diameter of the tubes was about 25 nm, the tube wall was smooth, and the thickness was uniform. In addition, a cross-sectional view of TiO_2_ NTAs is shown in Figure 3c, and we can observe that the thickness of the array structure is relatively uniform and its thickness is about 1.5 μm. In this way, the top-down nanotube array structure can generate a large specific surface area and expand the area of the p–n junction dissipation region, which provides favorable conditions for sufficient contact with ultraviolet light in the subsequent application of ultraviolet spot detectors. The bottom view of the sample is shown in Figure 3b, where it can be further observed that the TiO_2_ NTs have uniform and independent tube diameter sizes, smooth bottoms, and tight overall structures. Figure 3d shows the structure of the TiO_2_ NTA/PANI composite material. When the TiO_2_ NTAs were electrochemically composited with PANI, it could be observed that the mouth of the tubes, the tubes, and the inter-tube area were uniformly covered with PANI particles, showing an excellent composite effect.

Figure 4a shows the XRD pattern of TiO_2_ NTA/PANI composite nanomaterials. The TiO_2_ NTAs obtained under these conditions were binary, mixed, crystalline-type NTAs doped with rutile and anatase, and the contact interface between them forms an anatase/rutile (A/R) heterojunction. The intensity and number of peaks showed that the peaks were predominantly rutile peaks. Apart from the characteristic peaks of titanium flakes, there were no other impurity peaks in this sample. The Raman spectra of TiO_2_ NTA/PANI nanocomposites are shown in Figure 4b. Among them, the peaks at 141, 233, 443, and 611cm^−1^ are characteristic rutile TiO_2_ peaks [49], also indicating that the rutile type dominated the samples obtained by the above preparation method.

The intrinsic structural peaks of PANI can also be observed at 1196 cm^−1^ around the C-H plane bending vibration (benzene structure), 1293 cm^−1^ around the protonated structure, 1376 cm^−1^ around the C-N+ stretching vibration (semiquinone ring), 1569 cm^−1^ around the C=C stretching vibration (quinone structure), and 1621 cm^−1^ around the C-C stretching vibration in the benzene ring (benzene structure). A C-C stretching vibration in the benzene ring (benzene structure) occurs near 1621 cm^−1^ [46,50].

### 3.3. Optoelectronic Property Testing of TiO_2_ NTA/PANI Nanocomposites

The UV-Vis absorption spectra of TiO_2_ NTAs and TiO_2_ NTA/PANI nanocomposites are shown in Figure 5a. It can be seen that the absorption of pure TiO_2_ NTAs and TiO_2_/PANI nanocomposites in the UV region in the range of 200 nm~400 nm was significantly higher than that in the visible region of 400 nm~600 nm, especially after the combining TiO_2_ NTAs and PANI. The absorption of the composite material in the UV region increased by 20% compared to that of the pure TiO_2_ NTs, exhibiting a more effective detection of UV light. The detection effect of UV light was more significant.

Figure 5b shows the voltametric characterization curves of TiO_2_ NTA/PANI nanocomposites under UV irradiation at different wavelengths. The samples were irradiated under 365, 312, and 254 nm UV light and were subjected to photocurrent testing under dark conditions as a blank control. From this, it can be observed that the devices had obvious rectification characteristics, which proved that TiO_2_ effectively formed a p–n heterojunction after being combined with PANI using an electrochemical method. Meanwhile, it was not difficult to observe that its current under UV irradiation was higher than that in a dark environment; at the same voltage, the photocurrent increased with the increase in the wavelength of irradiated UV light and it had a better sensitivity to UV light.

Repeatability and the response speed are important factors when examining the performance of UV photodetector devices. The characteristic photocurrent–time response curves of pure TiO_2_ NTAs and TiO_2_ NTA/PANI nanocomposites are shown in Figure 6a, indicating a good stability and repeatability. We used the electrochemical workstation (CHI660D), selected amperometric i–t curve parameters, and set the initial voltage to 0 V, the sampling interval to 0.1 s, the running time to 400 s, the quiet time to 0 s, and the sensitivity to 1 × 10^−4^. Under the conditions of no applied voltage and darkness, it could be observed that there was almost no current generation for both samples. After turning on the UV light source and selecting an UV irradiation wavelength of 365 nm, the materials were in full contact with the UV light and both of them instantly generated a photocurrent in a very short time. When the current was stabilized and the UV light power supply was turned off, the current was then restored to the initial level in a very short time. The photocurrent generated by TiO_2_ NTA/PANI nanocomposites was three orders of magnitude larger than that of the pure TiO_2_ NTAs. As shown in Figure 6a,c, the response time of the pure TiO_2_ NTA device was 4.6 s/4.1 s when the light source was switched, while the response time of the TiO_2_ NTA/PANI nanocomposite device was 1.15 s/0.75 s, its photocurrent reached the desired level in a very short time, and its sensitivity to UV light and the response rate were much higher than those of the former. The characteristic photocurrent–time response curves of TiO_2_ NTA/PANI nanocomposites under three different wavelengths of UV irradiation (254, 312, and 365 nm) with an applied voltage of characteristic are shown in Figure 6b, from which we can observe that the TiO_2_ NTAs had a high sensitivity to UV light and a fast response rate after being effectively composited with PANI. At the same time, we can see that in the UV region, the longer the wavelength used to irradiate the sample, the larger the generated photocurrent.

In Figure 6c, we hypothesize that the heterojunction between pure TiO_2_ and the current collector is the reason that pure TiO_2_ materials also produce a photocurrent. In our test, the TiO_2_, Ti substrate, and FTO contacts make up the majority of the heterojunctions that may interact with photogenerated electrons. A faint photocurrent can form because FTO is a fluorine-doped SnO_2_ (tin dioxide) film that forms a very weak heterojunction at the point where it comes into contact with the TiO_2_ nanotube layer. Additionally, between the TiO_2_ nanotube layer and the substrate Ti sheet layer, there will be a metal/semiconductor metal oxide heterojunction that has the potential to create an integrated electric field [51,52,53]. A weak photocurrent is observed at 0 V bias in pure metal–oxide–semiconductor devices, which is consistent with many other similar device architectures. Furthermore, based on the SEM findings, we hypothesize that the TiO_2_ nanotube layer and the FTO layer have significantly less stratum areas than the TiO_2_ NTA/PANI device, resulting in a reduced built-in electric field. The built-in electric field of pure TiO_2_ is much weaker than the built-in electric field of TiO_2_ NTA/PANI devices, so the photocurrent is smaller when testing the photocurrent of pure TiO_2_ devices. In conclusion, even pure TiO_2_ devices have an internal electric field when testing the photocurrent, which can generate a photocurrent.

Responsivity (R) is a key parameter that determines the sensitivity of a photodetector device. The greater the R, the higher the sensitivity, expressed by the following formula [54]:R = ∆I/P

ΔI is the difference between the photocurrent and the dark current and P is the optical power. Under 365 nm light irradiation (light intensity is 310 μW/cm^2^), the effective irradiation area was about 1 cm^2^, and when the bias voltage was 0 V, the calculated R value of the TiO_2_ NTA device was 0.002 A/W. TiO_2_ NTA/PANI devices had R values of 0.026 A/W when irradiated with 365 nm light. Obviously, TiO_2_ NTA/PANI had a bigger R, indicating that the device had the best responsivity. When changing the wavelength of irradiating ultraviolet light, it can be calculated that the R value of TiO_2_ NTAs/PANI at 254 nm was 0.005 A/W at 254 nm and 0.008 A/W at 312 nm, both of which are less than the R value at 365 nm. In summary, the TiO_2_ NTA/PANI device had good spectral selection responsiveness, high detection rate, and responsivity.

Figure 7 shows the PL spectra of TiO_2_ NTAs and TiO_2_ NTA/PANI nanocomposites. After excitation with ultraviolet light of different wavelengths, pure TiO_2_ has a strong luminous intensity, which is a typical optical characteristic of TiO_2_ semiconductors and confirms that the prepared TiO_2_ has good photoexcitation properties. At the same time, it can be clearly observed from Figure 7 that when TiO_2_ NTAs were loaded with PANI, the luminous intensity was significantly weakened, indicating that the presence of PANI can transfer photogenerated electrons and thereby effectively inhibiting the recombination of photogenerated electron–hole pairs and improving the charge separation efficiency. At the same time, this also confirms the existence of the built-in electric field at the interface of TiO_2_ NTA and PANI.

The heterojunctions in the anatase/rutile binary mixed-crystal TiO_2_ NTAs prepared in this work can promote the transfer, separation, and migration of internal electrons and holes, which can effectively enhance the photosensitivity and photoresponsivity by suppressing carrier recombination and achieve better photoelectrochemical performances than those of the single-crystal TiO_2_ devices [55,56]. Meanwhile, the p–n heterojunction formed by electrochemically combining TiO_2_ NTAs with PANI can improve the utilization of UV light in UV point detector devices, which offers the possibility of improving the photoelectric efficiency, response speed, and UV sensitivity of the devices.

### 3.4. Internal Electron Transfer Mechanism of TiO_2_ NTA/PANI Materials

A space charge zone will arise at the interface as a result of electrons and holes being transported during the recombination of the two due to the various heights of the Fermi energy level. PANI is a p-type semiconductor that primarily conducts holes with a band gap of 2.80 eV and a Fermi level that is in close proximity to the valence band [57]. With a band gap of 3.2 eV and a Fermi level that is quite close to the conduction band, TiO_2_ is an n-type semiconductor that primarily conducts electrons [58]. Figure 8 shows a schematic representation of the energy band at the PANI/TiO_2_ nanocomposite’s PANI/TiO_2_ interface. As shown in the picture, when a p–n heterojunction between PANI and TiO_2_ is created, since PANI is primarily conducted by holes and TiO_2_ is primarily conducted by electrons, holes will transfer from TiO_2_ to PANI and electrons will pass from PANI to TiO_2_. A built-in electric field will be created by the transfer of electrons and holes. The built-in electric field runs in the opposite direction from where electrons are moving. The Fermi energy level at the interface of PANI and TiO_2_ will then be flat, as holes and electrons attain dynamic equilibrium. The energy band structure at the contact is twisted as a result of the creation of the internal electric field. The p–n heterojunction is created by the formation of an electron accumulation layer on the PANI side and an electron depletion layer on the TiO_2_ side [59]. Photovoltaic heterojunction UV photodetectors are built around this.

## 4. Conclusions

In this work, TiO_2_ NTA/PANI composite nanomaterials were successfully grown on titanium sheets by applying an anodic oxidation electrochemical method, and their photoresponsivity properties were further investigated. The findings demonstrated that a good p–n heterostructure evolved within the TiO_2_ NTA/PANI composites, which produced a clear rectification effect in their voltametric characteristic curves. They also demonstrated a high reaction rate, a strong UV photosensitivity, and good repeatability. This was primarily caused by the development of a p–n heterostructure, which prevented the combination of photogenerated electron–hole pairs and increased the device’s quantum efficiency and ability to use UV light. This provides a theoretical foundation and technical direction for the use of innovative UV photodetectors.

## Figures and Tables

**Figure 1 polymers-15-04400-f001:**
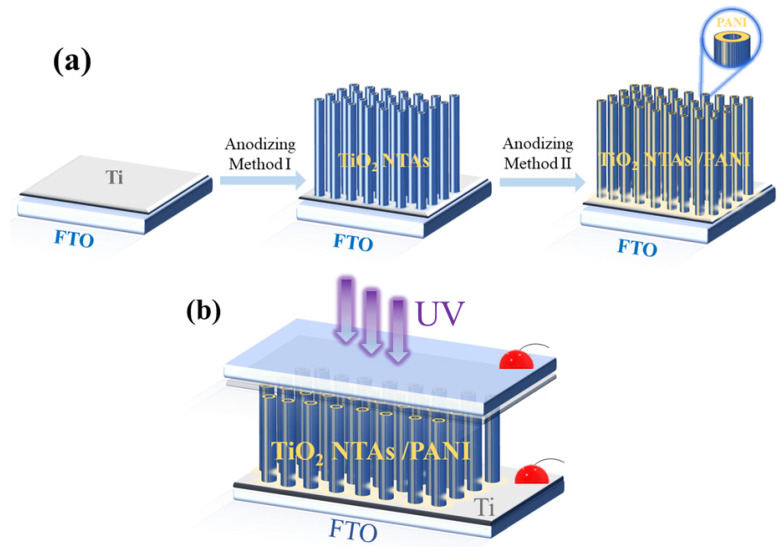
Schematic diagram of TiO_2_ NTA/PANI device preparation (**a**); device structure diagram (**b**).

**Figure 2 polymers-15-04400-f002:**
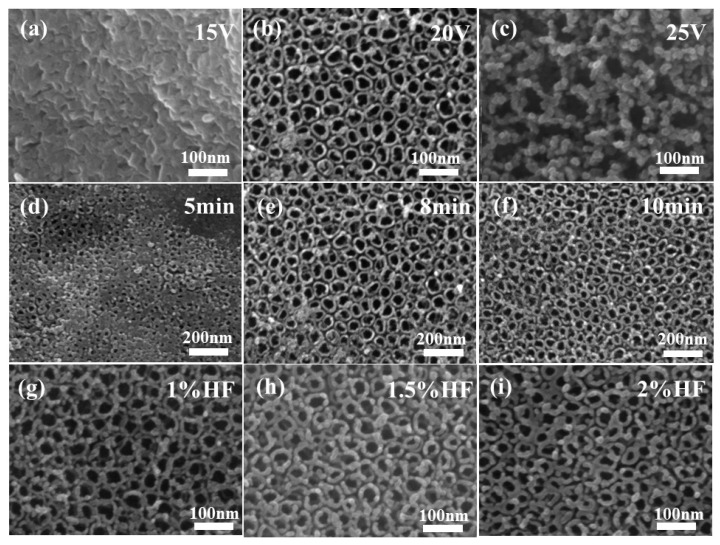
SEM diagram of TiO_2_ NTAs under different conditions: oxidation potential 15 V (**a**), 20 V (**b**), 25 V (**c**); oxidation time 5 min (**d**), 8 min (**e**), 10 min (**f**); electrolyte concentration 1% HF (**g**), 1.5% HF (**h**), 2% HF (**i**).

**Figure 3 polymers-15-04400-f003:**
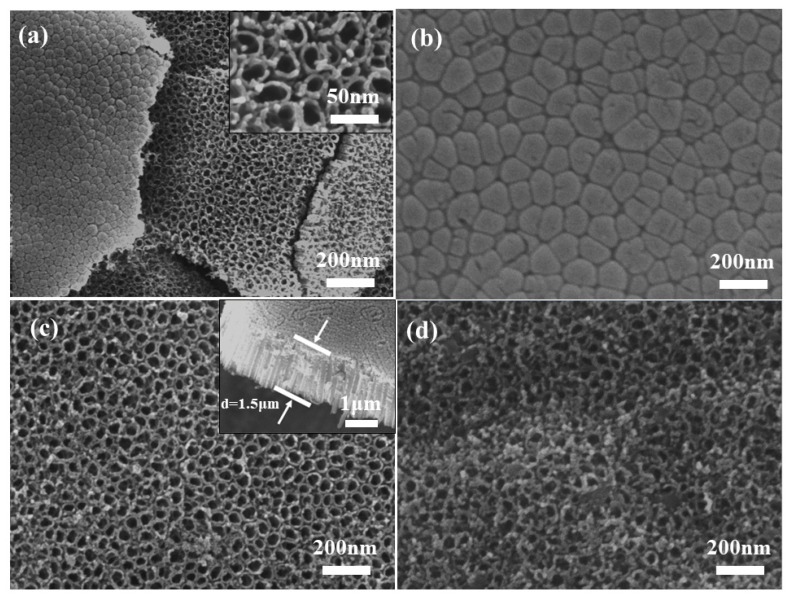
SEM images of TiO_2_ NTAs (**a**–**c**), TiO_2_ NTA/PANI nanocomposite (**d**).

**Figure 4 polymers-15-04400-f004:**
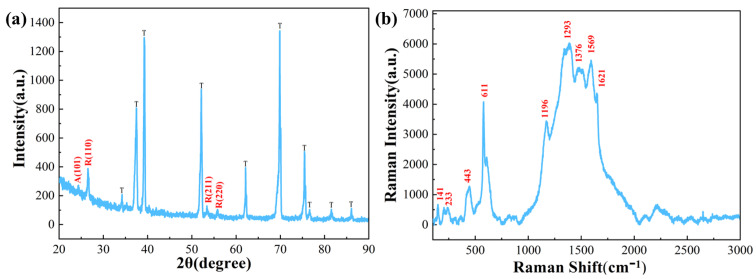
(**a**) XRD and (**b**) Raman spectroscopy of TiO_2_ NTAs.

**Figure 5 polymers-15-04400-f005:**
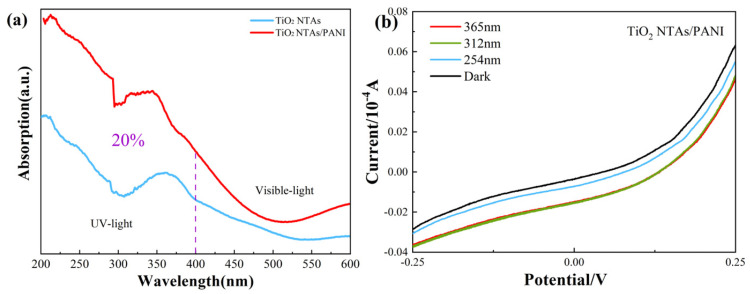
UV–vis light absorption spectra (**a**) and I–V curves (**b**) of TiO_2_ NTAs and TiO_2_ NTA/PANI nanocomposite.

**Figure 6 polymers-15-04400-f006:**
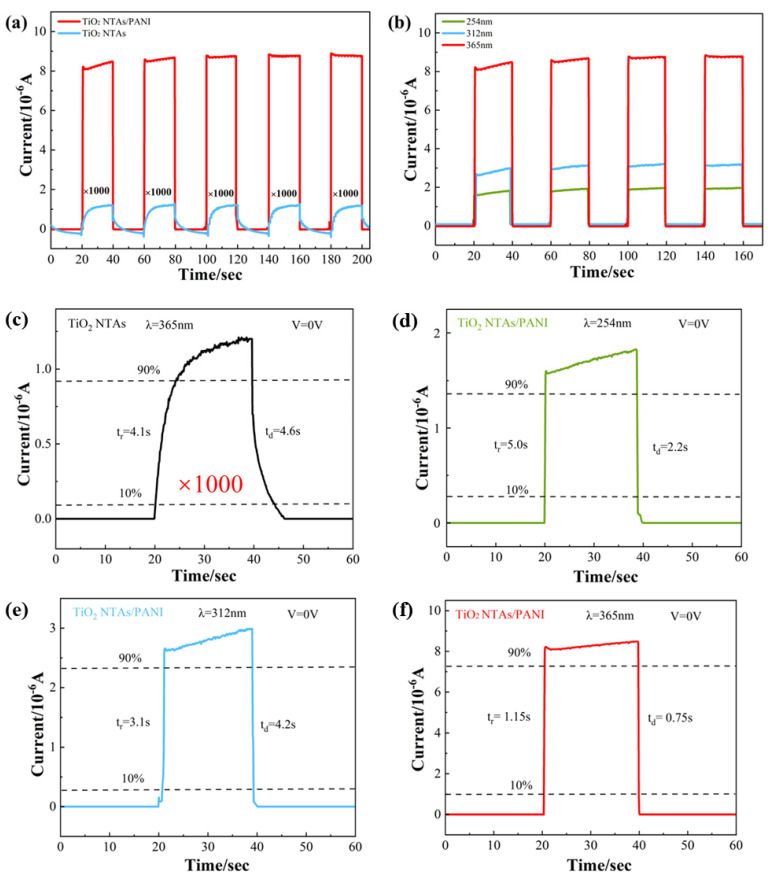
I–t curves of different devices at 0 V bias under 365nm UV light irradiation (**a**) and TiO_2_ NTA/PANI devices at different wavelengths (**b**). A sharp rise and fall in the enlarged portions of a 0–60 s range correspond to the on-state and off-state at a bias voltage of 0 V for (**c**) TiO_2_ NTAs, 365 nm and TiO_2_ NTA/PANI (**d**) 254 nm (**e**) 312 nm (**f**) 365 nm.

**Figure 7 polymers-15-04400-f007:**
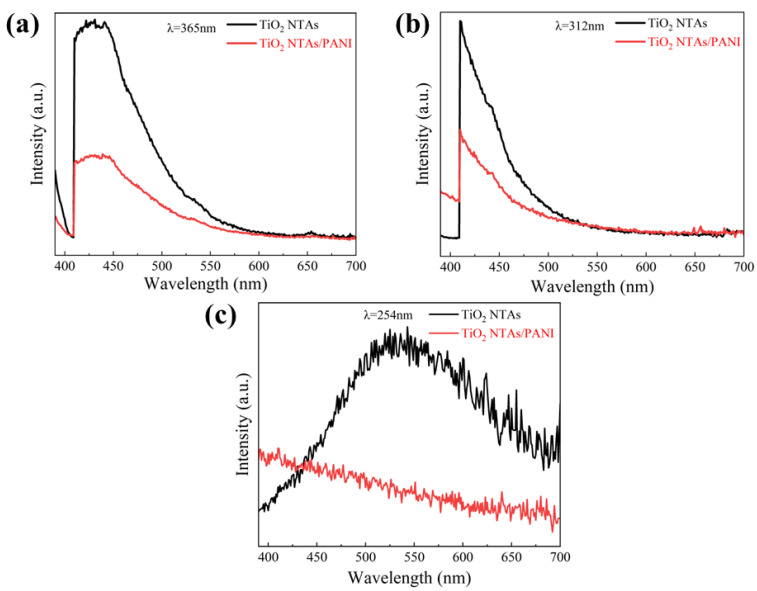
PL spectra of TiO_2_ and PANI/TiO_2_ under different excitation wavelengths of UV light: (**a**) 365 nm, (**b**) 312 nm, (**c**) 254 nm.

**Figure 8 polymers-15-04400-f008:**
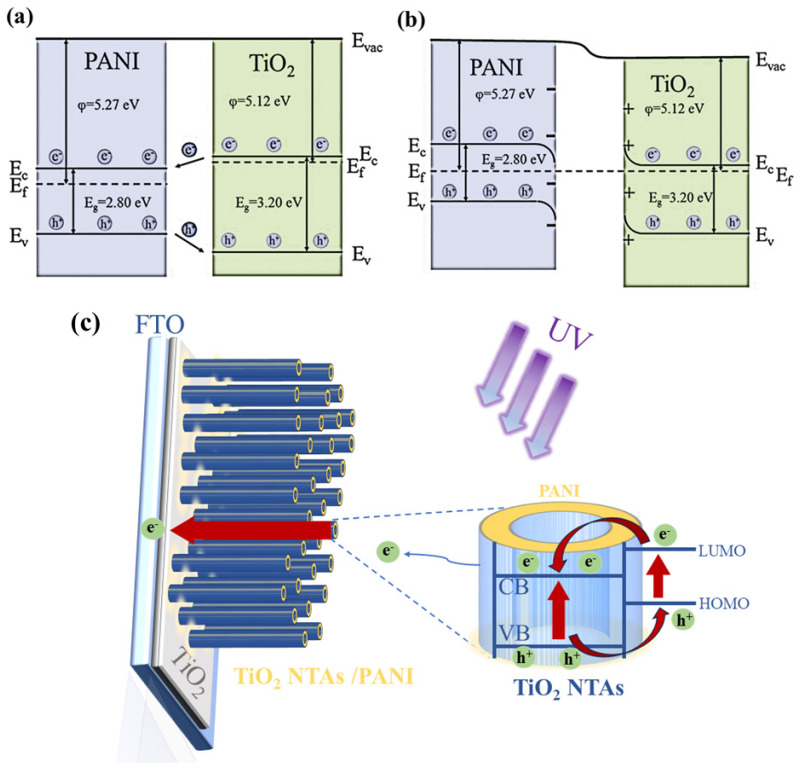
The contact potential differences of (**a**) PANI and (**b**) TiO_2_. (**c**) Schematic diagram of interface electron transfer.

## Data Availability

The data presented in this study are available on request from the corresponding author.

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
