# Peer review of "Preparation and Photovoltaic Performance of a Composite TiO2 Nanotube Array/Polyaniline UV Photodetector"

_polymers, 2023, doi:10.3390/polym15224400_

Round 1

Reviewer 1 Report

In this study, the authors prepared and investigated the TiO2 NTAs/PANI for potential UV sensor applications. The article appears to be interesting but can be reconsidered for publication after a major revision. 

1) The authors stated that the thickness is uniform - please provide the corresponding cross-sectional image and thickness value.

2) XRD study. Analyze the % ratio between anatase and rutile TiO2 phases. What was the amount of PANI in TiO2? determine using the TGA analysis. 

3) According to the UV-Vis study, the TiO2 NTAs/PANI sample is still responsive to visible light, please provide I-V curves under visible light for comparison. 

4) Better charge separation in TiO2 NTAs/PANI samples should be confirmed by PL analysis. 

5) Please investigate whether intensity in I-t depends on the wavelength intensity. How the temperature and humidity affect the I-t curves.

The manuscript can be polished one more time. 

Author Response

Reviewer #1:

In this study, the authors prepared and investigated the TiO2 NTAs/PANI for potential UV sensor applications. The article appears to be interesting but can be reconsidered for publication after a major revision.

Answer: Thank you for your comments. We feel grateful for your professional reviews work on our article. As you are concerned, there are some questions need to be addressed. According to your nice suggestions, we have made extensive corrections to our previous manuscript. The detailed corrections are listed below. Thanks again for your constructive comments.

1) The authors stated that the thickness is uniform - please provide the corresponding cross-sectional image and thickness value.

Answer: Thank you for your comments. According to your suggestion, in order to better represent the uniformity of TiO2 NTAs, we added a cross-sectional SEM diagram of the material, and it is clear from Figure.R1 that the material thickness distribution is relatively uniform, and its thickness value is about 1.5 microns.

Figure.R1 SEM of cross-section of TiO2 NTAs.

2) XRD study. Analyze the % ratio between anatase and rutile TiO2 phases. What was the amount of PANI in TiO2? determine using the TGA analysis.

Answer: Thank you for your professional comments. The TiO2 nanotube array obtained in this work is a rutile and anatase-doped crystal form, and it can be seen from the peak intensity and quantity that the rutile type characteristic peak is the main one. Except for the characteristic peaks of the titanium sheet, there are no other impurity peaks in this sample. Through the calculation of the peak area of TiO2 anatase type and rutile type, it can be concluded that the proportion of anatase type is about 21%, and the proportion of rutile type is 79%. For TGA analysis, because the research space of this work is limited, there is no quantitative analysis of the content of polyaniline, and the effect of polyaniline thickness and quality on the material cannot be further studied, referring to the relevant literature of this work, Meng Yang et al. [1] TGA analysis of TiO2 NTAs/PANI composites proves that PANI has less content than TiO2, accounting for about 10%. Thanks again for your valuable comments.

3) According to the UV-Vis study, the TiO2 NTAs/PANI sample is still responsive to visible light, please provide I-V curves under visible light for comparison.

Answer: Thank you for your comments. Polyaniline (PANI) is a conductive polymer material with a high absorption coefficient in the visible range. TiO2 can only absorb ultraviolet light in the solar spectrum, but cannot harness the energy of visible and near-infrared light. In this work, TiO2 NTAs/PANI materials still react under visible light because of visible light absorption caused by the diffraction of polyaniline. Due to the limited space of this work, the main research is the response of the device to ultraviolet light, and the I-V curve performance of TiO2 NTAs/PANI materials in visible light has not been further explored. Thank you for this very constructive suggestion, the response to visible light will be the direction of our further exploration in the future. Hopefully, the explanation we give will relieve your doubts. Thanks again for your comments.

4) Better charge separation in TiO2 NTAs/PANI samples should be confirmed by PL analysis.

Answer: Thank you for your comments. Based on your recommendations, we have supplemented PL testing on TiO2 NTAs/PANI materials with detailed analysis in the manuscript. The test results show that pure TiO2 has strong luminous intensity, which belongs to the typical optical properties of TiO2 semiconductors, and confirms that the prepared TiO2 has good photoexcitation properties. After loading PANI, the luminous intensity was significantly weakened, indicating that the presence of PANI could transfer photogenerated electrons and effectively inhibit the recombination of photogenerated electron-hole pairs. This also confirms the existence of a built-in electric field at the TiO2/PANI interface. We hope you are satisfied with what we have added. Thank you again for your very professional comments.

Figure.7 PL spectra of TiO2 and PANI/TiO2 under different excitation wavelengths of UV light (a)365nm(b)312nm(c)254nm.

5) Please investigate whether intensity in I-t depends on the wavelength intensity. How the temperature and humidity affect the I-t curves.

Answer: Thank you for your comments. This work mainly explores the influence of different wavelengths of ultraviolet light on the performance of photodetector devices, focusing on the influence of illumination. Due to the limited space of this work, the effects of temperature and humidity on the I-t curves of TiO2 NTAs/PANI composites cannot be further explored. We have reviewed a large number of relevant literature, among which, for the effects of environmental factors such as temperature and humidity, in "Resistive room temperature DMA gas sensor based on the forest-like unusual n-type PANI/TiO2 nanocomposites", Meng Yang et al [1] showed in detail that TiO2 NTAs/PANI conforming materials have good long-term temperature stability and moisture resistance. At the same time, in "Photodegradation of Textile Pollutants by Nanocomposite Membranes of Polyvinylidene Fluoride Integrated with Polyaniline-Titanium Dioxide Nanotubes", Hafiza Hifza Nawaz et al [2]. It also proves that the material is minimally affected by ambient humidity. In summary, the influence of temperature and humidity on the performance of TiO2 NTAs/PANI composites as ultraviolet photodetectors in this work is small and can be ignored. We sincerely hope that the answers would meet your requirements. Thanks again for your professional comments.

Comments on the Quality of English Language:The manuscript can be polished one more time.

Answer: Thank you for your professional comments. We feel sorry for our poor writings, however, we do invite a friend of us who is a native English speaker from the USA to help polish our article. We have tried our best to polish the language in the revised manuscript. And we hope the revised manuscript could be acceptable for you. Thanks again for your comments.

Reviewer 2 Report

This manuscript focuses on the development of a composite UV photodetector using TiO2/PANI nanotube arrays. While the study holds some potential significance for UV photodetection applications, the current version of the manuscript requires a major revision to address several crucial aspects, ensuring clarity, comprehensiveness, and scientific rigor.

1. Abstract, Line 17: what’s the meaning of the term "complexation of photogenerated electron-hole pairs" in the context of this study?

2. Introduction: The manuscript lacks a comprehensive review of related works, and the distinctiveness of this study from previous research is not adequately highlighted.

3. What’s the specific reason for selecting TiO2 nanotube arrays (NTAs) over TiO2 films?

4. The manuscript should include a detailed band diagram of the TiO2/PANI composite, illustrating energy levels and explaining the charge transfer mechanism within the material.

5. The authors need to clarify why a transparent conducting layer (FTO) was used as the substrate, especially considering the presence of a Ti layer as the bottom electrode.

6. Page 5, Line 161: The statement made here is inaccurate. As the authors claim a photovoltaic effect in the UV detector, it is important to note that the effective light sensing will take place at the depletion region of the p-n heterostructure, not at the surface of TiO2.

7. Page 6, line 193: A 20% increment in absorbance lacks significance without the inclusion of film thickness information, as absorbance is dependent on thickness.

8. Figure 5b: The absence of response at reverse bias in a photodiode-based device requires an in-depth explanation.

9. Calculations and discussions of the responsivity and detectivity should be included to provide a comprehensive evaluation of its performance.

10. How was the dynamic response (Figure 6) measured at a 0 V bias? Please provide detailed information in the experimental section.

11. Why is the photocurrent at the 365nm wavelength higher than at 254nm and 312nm, which contradicts the absorption spectrum?

English requires extensive polishing to enhance clarity and coherence.

Author Response

Reviewer #2:

This manuscript focuses on the development of a composite UV photodetector using TiO2/PANI nanotube arrays. While the study holds some potential significance for UV photodetection applications, the current version of the manuscript requires a major revision to address several crucial aspects, ensuring clarity, comprehensiveness, and scientific rigor.

Answer: Thank you for your comments. We feel grateful for your professional reviews work on our article. As you are concerned, there are some questions need to be addressed. According to your nice suggestions, we have made extensive corrections to our previous manuscript. The detailed corrections are listed below. Thank you very much.

  1. Abstract, Line 17: what’s the meaning of the term "complexation of photogenerated electron-hole pairs" in the context of this study?

Answer: Thank you for your comments. In the abstract, "complexation of photogenerated electron-hole pairs" refers to a phenomenon in which the electron-hole formed by the semiconductor under illumination recombines with another electron so that the original electron and electron-hole disappear and release energy. We know that photodetectors play a role in converting optical signals into electrical signals in practical applications, mainly based on the principle of the photovoltaic effect or photoconductivity effect of materials. In general, the basic working mechanism of photodetectors includes three processes: (1) the generation of photogenerated carriers under illumination; (2) the Separation of photogenerated electrons and holes; (3) carrier diffusion or drift forms a current. Among them, photogenerated electrons and hole separation are the key to realizing photoelectric detection technology. In semiconductors, when an electron transitions from the valence band to the conduction band, it leaves an electron hole. This energy can be released in the form of heat or, in some cases, light, forming a semiconductor luminescence phenomenon. However, due to the small distance between photogenerated electrons and holes inside the semiconductor and the electrostatic force of Coulomb, it is easy to recombine during transmission, thereby inactivating carriers and seriously affecting the detection performance of the device. Therefore, the recombination of photogenerated electrons and holes should be avoided. We sincerely hope that the answers would meet your requirements. Thanks again for your professional comments.

  1. Introduction: The manuscript lacks a comprehensive review of related works, and the distinctiveness of this study from previous research is not adequately highlighted

Answer: Thank you for your comments. Based on your constructive suggestions, we have supplemented the introduction with a detailed description of the relevant research work in this field, as well as further highlighting the differences between this work and other studies, highlighting the advantages of this work. We have revised the expressions and sincerely hope that the answers would meet your requirements. Thanks again for your professional comments.

Original:

In recent years, UV photodetectors have gradually become a hot spot in the field of scientific research [1,2]. UV photodetectors in the mid-ultraviolet and near-ultraviolet wavelength bands have widely used in major fields, such as military and defense [3], biomedicine [4,5], environmental monitoring [6], and light wave communication [7] due to their excellent performance in high sensitivity, wide linear range, and good stability. Compared with traditional infrared and other types of photodetectors, UV photodetectors show remarkable advantages, such as excellent photosensitivity and accuracy, strong concealment, high military value, and fast presentation of detection results [8-10]. Despite the numerous researches conducted on UV photodetectors, there are still many unsolved problems, such as the difficulty and high cost of preparing wide-band semiconductor arrays, the poor stability of the material itself, and the in-sufficient performance of the devices [11-13].

At present, a variety of semiconductor materials with different bandwidths have been used as substrate materials for UV photodetectors due to their own advantages [14-20]. TiO2 has the advantages of good chemical stability, low toxicity, environmental friendliness, and it is one of the most widely used wide bandwidth semiconductors nowadays [21-23]. Among them, the rutile TiO2 has a bandwidth of about 3.0 eV, and the anatase TiO2 has a bandwidth of about 3.2 eV [24]. Because of the valence band energy difference between the two crystal phases, the heterojunction constructed by combining the two can reduce the forbidden bandwidth, resulting in a mixed crystal effect [25], which can promote the separation of photogenerated electrons and hole pairs to a certain extent. The above excellent optoelectronic properties have attracted much attention in ultraviolet photodetection [26-27]. Among them, TiO2 NTAs are highly ordered in structure, which can provide numerous charge transport channels, a larger specific surface area, and more interaction sites with UV light, providing great possibilities for the design and manufacture of higher-performance and high-er-efficiency UV photodetectors [28-32]. However, TiO2 as an n-type metal oxide sem-iconductor has a very high internal charge complex rate, and TiO2-based photodetec-tors require an external voltage to provide potential difference excitation for proper operation, which leads to large device size and severely limits their application in real life [33].

Polyaniline (PANI) is a common p-type conductive polymer with excellent elec-trical conductivity and electrochemical properties. Meanwhile, it has the advantages of high capacitance, high carrier mobility, high light absorption coefficient, ease of syn-thesis, and good environmental friendliness [34-37]. It is well known that the built-in electric field of p-n heterojunction can induce photoelectrons and holes to be trans-ferred in opposite directions, which can achieve the purpose of inhibiting photogener-ated carriers from compounding with each other [38,39]. Therefore, the p-n hetero-junction can be constructed by effectively compounding the n-type metal oxide semi-conductor TiO2 with the p-type conductive polymer PANI [40-42], and the potential difference automatically formed inside the heterojunction can provide the impetus for the separation of photogenerated electron-hole pairs and the formation of photocurrent, which can make the UV photodetectors have excellent UV sensitivity [43-45].

In this work, it was proposed that the composite of TiO2 NTAs with the P-type conjugated conducting polymer PANI can form a good p-n heterojunction interface, which was conducive to accelerating the separation of photogenerated electron-hole pairs, suppressing carrier complexation, and improving the quantum efficiency, so as to further improve the response speed, sensitivity, and stability of the UV photodetector.

Revised:

In recent years, UV photodetectors have gradually become a hot spot in the field of scientific research [1]. UV photodetectors in the middle and near-ultraviolet bands are widely used in military defense [3], biomedicine [4], environmental monitoring [5,6], and optical wave communication [7] because of their excellent performance such as high sensitivity, wide linear range, and good stability. The use of ultraviolet rays in sunlight or flames for ozone holes and fire monitoring has gradually become the focus of research. Compared with traditional infrared and other types of photodetectors, UV photodetectors show excellent advantages, such as excellent photosensitivity and accuracy, strong concealment, high military value, and can quickly present detection results [8-10]. Although there have been countless studies on UV photodetectors, there are still many unsolved problems, such as the difficulty of preparing wide bandgap semiconductor arrays, high preparation costs, poor stability of the material itself, and insufficient device performance [11-13]. To meet the needs of various fields, high-performance, high-stability, high-reliability UV photodetectors, especially UV detectors that can work under extreme and harsh conditions, are extremely important, and the reliability and accuracy of UV photodetectors have a crucial impact on practical applications.

At present, a variety of semiconductor materials with different bandwidths have been used as substrate materials for UV photodetectors due to their own advantages [14-20]. TiO2 has the advantages of good chemical stability, low toxicity, environmental friendliness, and it is one of the most widely used wide bandwidth semiconductors nowadays [21-23]. Among them, the rutile TiO2 has a bandwidth of about 3.0 eV, and the anatase TiO2 has a bandwidth of about 3.2 eV [24]. Because of the valence band energy difference between the two crystal phases, the heterojunction constructed by combining the two can reduce the forbidden bandwidth, resulting in a mixed crystal effect [25], which can promote the separation of photogenerated electrons and hole pairs to a certain extent. The above excellent optoelectronic properties have attracted much attention in ultraviolet photodetection [26-27]. Among them, TiO2 NTAs are highly ordered in structure, which can provide numerous charge transport channels, a larger specific surface area, and more interaction sites with UV light, providing great possibilities for the design and manufacture of higher-performance and high-er-efficiency UV photodetectors [28-32]. However, TiO2 as an n-type metal oxide sem-iconductor has a very high internal charge complex rate, and TiO2-based photodetec-tors require an external voltage to provide potential difference excitation for proper operation, which leads to large device size and severely limits their application in real life [33].

Polyaniline (PANI) is a common p-type conductive polymer with excellent elec-trical conductivity and electrochemical properties. Meanwhile, it has the advantages of high capacitance, high carrier mobility, high light absorption coefficient, ease of syn-thesis, and good environmental friendliness [34-37]. It is well known that the built-in electric field of p-n heterojunction can induce photoelectrons and holes to be trans-ferred in opposite directions, which can achieve the purpose of inhibiting photogener-ated carriers from compounding with each other [38,39]. Therefore, the p-n hetero-junction can be constructed by effectively compounding the n-type metal oxide semi-conductor TiO2 with the p-type conductive polymer PANI [40-42], and the potential difference automatically formed inside the heterojunction can provide the impetus for the separation of photogenerated electron-hole pairs and the formation of photocurrent, which can make the UV photodetectors have excellent UV sensitivity [43-45].

The microstructure of TiO2 nanotube arrays (NTAs) is neatly arranged nanotubes and fixed on the titanium substrate, which has a large specific surface area, good stability, and easy recyclability. However, due to the large band gap, response to ultraviolet light, and high electron-hole pair recombination rate of TiO2, it is necessary to improve the performance of TiO2 materials by modifying doping. At present, many researchers have cleverly compounded TiO2 NTAs with conductive polymer PANI to obtain countless materials with excellent performance. Wang et al. [47] prepared a TiO2 nanotube array photoelectrode (PANI/TiO2) wrapped by polyaniline (PANI) nanosheets by simple anodizing method and vacuum-assisted impregnation method. The results show that when the polyaniline nanosheets are tightly wrapped in TiO2 nanotubes, they have higher light response and charge separation efficiency than pure TiO2. Among them, the photocurrent density of PANI/TiO2-0.1 is 0.73 times that of TiO2. On PANI/TiO2-0.1, the degradation activity of PEC p-phenol was 27.7% higher than that of TiO2. It was proved that the synergistic effect of photocatalysis and electrocatalysis was generated on the polyaniline/TiO2 photoelectrode. Chen et al. [51] prepared a PANI/H-TiO2 nanotube array composite electrode by constant voltage deposition in electrochemical deposition. They first obtained the titanium dioxide nanotube array by secondary anodizing of the metal titanium sheet, then calcined and hydrogenated it in a hydrogen environment, and finally placed the hydrogenated sample in acetone solution containing sulfuric acid and aniline, and deposited polyaniline on the titanium dioxide nanotube array by constant voltage method to obtain a PANI/H-TiO2 nanotube array composite electrode. They measured the solution resistance and charge transfer resistance of 0.1554 and 2.723 V cm2, respectively, indicating that PANI/H-TiO2 nanotubes have better electron transport ability and high charge transfer rate. Xie et al. [52] prepared a PANI-TiO2 nanotube array composite electrode using the potentiostatic potential polymerization method in electrochemistry. First, the anodized titanium dioxide nanotube array is immersed in a hydrochloric acid solution containing aniline monomer, stirred, and fully infiltrated, and then polyaniline is deposited on the nanotubes by potentiostat method (voltage 0.9 V) in the same solution to obtain a composite electrode. They observed that TiO2 nanotubes were wrapped with disordered polyaniline nanowire arrays, with unique microstructure, high specific surface area, fast ion diffusion path, and long-term cycle stability, and the composite showed excellent electrochemical performance. It can be seen that after the recombination of TiO2 nanotube array and PANI, the improvement of specific surface area, stability, and charge separation efficiency of the material is extraordinary, and it has great development prospects in the field of photoelectric detection materials in the future.

Based on the work of other researchers in this field, this work analyzes the current status of photodetectors and proposes to use a simpler two-step electrochemical method to composite TiO2 nanotube arrays with P-type conjugated conductive poly-mer PANI, to further expand the spectral response range of the material and increase the device's absorption of ultraviolet light, At the same time, a good p-n heterojunction interface can be formed inside the device, which is conducive to accelerating the sepa-ration of photogenerated electron hole pairs, suppressing carrier recombination, effec-tively improving quantum efficiency, and further improving the response speed, sen-sitivity, and stability of ultraviolet photodetectors.

  1. What’s the specific reason for selecting TiO2 nanotube arrays (NTAs) over TiO2 films?

Answer: Thank you for your comments. For the two different structures of the materials you mentioned, TiO2 nanotube arrays (NTAs) have better-photogenerated charge separation and transport characteristics than TiO2 films and efficient charge collection capabilities. TiO2 nanotube array is a top-down three-dimensional structure, which is widely used in ultraviolet photoelectric detection because of its highly ordered charge transport channel and large specific surface area, especially its low light-induced charge binding ability [3]. Compared with TiO2 particles and thin films, TiO2 nanotube arrays have better photoelectrochemical performance and photoconversion efficiency. In particular, the highly ordered nanoarray structure of TiO2 nanotube arrays can expose a large number of advantageous crystal planes that are conducive to charge separation, enhance light scattering, and increase the number of carriers generated by light. The carriers can perform long-distance rapid migration at the axial distance of the nanowire so that the photogenerated electrons can be rapidly exported, which reduces the probability of recombination of photogenerated electron-hole pairs, so a TiO2 nanotube array with better performance was selected in this work. We sincerely hope that the answers would meet your requirements. Thanks again for your professional comments.

  1. The manuscript should include a detailed band diagram of the TiO2/PANI composite, illustrating energy levels and explaining the charge transfer mechanism within the material.

Answer: Thank you for your professional comments. Based on your suggestions, we have supplemented the detailed band diagram of the TiO2 NTAs/PANI composite, as shown in Figure 8. We will further explain the mechanism of the material. The photovoltaic effect can be simply expressed as the device absorbing photons under light to excite the electron-hole pair, which was separated by the internal electric field to generate a voltage, thereby forming a photocurrent. Some interfaces can establish internal electric fields, such as Schottky junctions or PN junctions. In the darkfield, the device exhibits nonlinear I-V characteristics due to the presence of the PN junction, and the current increases exponentially with voltage. When illuminated, electron-hole pairs that absorb photon excitation can be separated by a built-in electric field without an external voltage, increasing photocurrent. At this time, the reverse voltage bias was added to the device, the photogenerated carriers moved in the opposite direction, the reverse current increased and the photon energy was converted into electrical energy through the photovoltaic effect. This mechanism can make the photodetector work under extremely low power consumption, achieve self-power, work without an external energy supply, have strong outdoor working ability, and have high practicality.

The TiO2 NTAs/PANI composite ultraviolet photodetectors prepared in this work belong to PN junction photodetectors. PN junction is formed by contact between P-type semiconductor material and N-type semiconductor material, The two materials are in a separate state before the formation of contact, the position of the Fermi level in the P-type semiconductor is close to the edge of the valence band, the Fermi level in the N-type semiconductor is close to the edge of the conduction band, when the two types of materials contact to form a PN junction, the Fermi level of the P-type material located at the edge of the valence band will continue to move up, at the same time, the Fermi level of the N-type material located at the edge of the conduction band continues to move down until the Fermi level of the two materials is in the same position. Changes in the position of the conduction band and valence band are accompanied by the bending of the energy band. At this point, the PN junction is in equilibrium and has a uniform Fermi level. From the aspect of carriers, most of the carriers in P-type materials are holes, and most of the carriers in N-type materials are electrons, when the two materials are in contact, due to the difference in carrier concentration, the electrons in N-type materials will diffuse into P-type, and the electrons of N-type materials spread in the opposite direction to the hole, and the uncompensated area left by the electron and hole diffusion will form a built-in electric field, and the built-in electric field will drive the carrier drift, and the direction of drift is just opposite to the direction of diffusion. It means that the formation of the built-in electric field prevents the diffusion of carriers, and there is diffusion and drift inside the PN junction at the same time until the two motions are equilibrium so that the static carrier flow is zero and the internal dynamic equilibrium is reached. Finally, when the PN junction is subjected to optical radiation, the energy of the photon is transferred to the carrier, producing a photogenerated carrier, that is, a photogenerated electron-hole pair, under the action of the electric field, the electron and hole drift to the N region and P region respectively, and the directional drift phenomenon of the photogenerated carrier produces a photocurrent. This is the basic principle of PN junction photodetectors. Compared with pure TiO2, the improvement of the electrochemical performance of TiO2 NTAs/PANI composites is attributed to the good synergistic effect of the components, which significantly reduces the bandgap energy and charge transfer resistance. We sincerely hope that the answers would meet your requirements. Thanks again for your professional comments.

Figure.8 The contact potential differences of (a) PANI and (b) TiO2.

  1. The authors need to clarify why a transparent conducting layer (FTO) was used as the substrate, especially considering the presence of a Ti layer as the bottom electrode.

Answer: Thank you for your professional comments. From the composition of ultraviolet photodetectors, photovoltaic detectors can be composed of a transparent conductive substrate, a layer of photosensitive material, and an opaque bottom metal electrode. Generally, transparent conductive substrates use indium tin oxide conductive glass (ITO) or fluorine-doped tin oxide conductive glass (FTO). As a carrier for other materials, light enters and transmits the collected photoelectrons to the outer circuit. FTO was selected as the transparent conductive substrate in this work to increase the ultraviolet light absorption of the device and assist TiO2 NTAs/PANI composites to form a complete UV photodetector device, to achieve excellent and stable light response performance of the device. Titanium sheet is part of the photosensitive material, which is the key basis for the growth of TiO2 NTAs, which is a template for uniform polymerization of aniline, and the improvement of the performance of TiO2 NTAs/PANI composites is attributed to the good synergistic effect of the components, thereby significantly reducing the bandgap energy and charge transfer resistance. The effective combination of the above parts forms a UV photodetector with a fast detection rate, high sensitivity, and self-power supply. We sincerely hope that the answers would meet your requirements. Thanks again for your professional comments.

  1. Page 5, Line 161: The statement made here is inaccurate. As the authors claim a photovoltaic effect in the UV detector, it is important to note that the effective light sensing will take place at the depletion region of the p-n heterostructure, not at the surface of TiO2.

Answer: Thank you for your professional comments. By your suggestion, and in keeping with scientific rigor, we have further revised the description in this section to ensure the accuracy of this work statement. We have revised the expressions and sincerely hope that the answers would meet your requirements. Thanks again for your professional comments.

Original:

Figure 3(a,c) showed the top view of the TiO2 NTAs, which can be seen that the TiO2 NTAs were perpendicular to the substrate of the titanium sheet, and each one of them was separate and independent with a relatively uniform size, and the diameter of the tubes was about 25 nm, the tube wall was smooth, the thickness was uniform, and the growth was good and a large specific surface area can be generated, which can provide a favorable condition for the subsequent application of the UV point detector with sufficient contact with the UV light.

Revised:

Figure 3(a,c) showed the top view of the TiO2 NTAs, which can be seen that the TiO2 NTAs were perpendicular to the substrate of the titanium sheet, and each one of them was separate and independent with a relatively uniform size, and the diameter of the tubes was about 25 nm, the tube wall was smooth, the thickness was uniform. In addition, a cross-sectional view of TiO2 NTAs was inserted in Figure 3(c), and we can observe that the thickness of the array structure is relatively uniform, and its thickness value is about 1.5μm. In this way, the top-down nanotube array structure can generate a large specific surface area and expand the area of the p-n junction dissipation region, which provides favorable conditions for sufficient contact with ultraviolet light in the subsequent application of ultraviolet spot detectors.

  1. Page 6, line 193: A 20% increment in absorbance lacks significance without the inclusion of film thickness information, as absorbance is dependent on thickness.

Answer: Thank you for your comments. On page 6, line 193, this work compares the thickness of the two materials based on the same, and the thickness value is about 0.1mm, which can ensure the better light transmittance of the material. We sincerely hope that the answers would meet your requirements. Thanks again for your valuable comments.

  1. Figure 5b: The absence of response at reverse bias in a photodiode-based device requires an in-depth explanation.

Answer: Thank you for your comments. Figure 5b mainly confirms that TiO2 NTAs/PANI have the rectification effect of semiconductor diode PN junction. When the PN junction is positively biased, the voltage rises rapidly after a certain value, and the PN junction is on, and when the PN junction is reversed, due to the presence of the built-in electric field, the current passing through is extremely small, which is the result shown on the left side in Figure 5b. When the reverse bias voltage is added to a certain extent, the current index increases, and the change in voltage is very small, which is called reverse breakdown. We sincerely hope that the answers would meet your requirements. Thanks again for your valuable comments.

  1. Calculations and discussions of the responsivity and detectivity should be included to provide a comprehensive evaluation of its performance.

Answer: Thank you for your professional comments. Responsivity (R) is a key parameter that determines the sensitivity of a photodetector device. The greater the R, the higher the sensitivity, expressed by the following formula [4]:

R = ∆I/PS

ΔI is the difference between the photocurrent and the dark current, P is the optical power, and S is the effective irradiation area. Under 365 nm light irradiation (light intensity is 310 μW/cm2), the effective irradiation area was about 1 cm2, and when the bias voltage was 0V, the R-value of the calculated TiO2 NTAs device was 0.002A/W. TiO2 NTAs/PANI devices had R values of 0.026 A/W when irradiated with 365 nm light. Obviously, TiO2 NTAs/PANI had the bigger R, indicating that the device had the best responsivity. When changing the wavelength of irradiated ultraviolet light, it can be calculated that the R-value of TiO2 NTAs/PANI at 254nm was 0.005A/W at 254nm and 0.008A/W at 312nm, both of which were less than the R-value at 365nm. In addition, as shown in Figure R2, in the process of light source switching switch, the response time of pure TiO2 nanotube array devices was (4.6s/4.1s), while the response time of TiO2 NTAs/PANI nanocomposite devices was (1.15s/0.75s), and its photocurrent can reach the ideal level in a very short time, and its sensitivity and response rate to ultraviolet light was much higher than the former. In summary, TiO2 NTAs/PANI had good spectral selection responsiveness, high detection rate, and responsivity. We sincerely hope that the answers would meet your requirements. Thanks again for your professional comments.

Figure.R2 A sharp rise and decay in the enlarged portions of a 0–60 s range corresponding to the on-state and off-state at a bias voltage of 0V for (a)TiO2 NTAs,365nm and TiO2 NTAs/PANI (b)254nm(c)312nm(d)365nm

  1. How was the dynamic response (Figure 6) measured at a 0 V bias? Please provide detailed information in the experimental section.

Answer: Thank you for your professional comments. In accordance with your suggestions, we have supplemented the description of the experimental test section to ensure the completeness of this work discussion. We have revised the expressions and sincerely hope that the answers would meet your requirements. Thanks again for your professional comments.

Original:

Under the condition of zero applied voltage and darkness, it can be observed that there was almost no current generation for both.

Revised:

In this work, the Amperometric i-t Curve Parameters were selected using the electrochemical workstation (CHI660D), and the initial voltage was set to 0V, the sampling interval was 0.1 seconds, the running time was 400 seconds, the quiet time was 0 seconds, and the sensitivity was 1e-4. With zero applied voltage and dark conditions, almost no current generation can be observed in both.

  1. Why is the photocurrent at the 365nm wavelength higher than at 254nm and 312nm, which contradicts the absorption spectrum?

Answer: Thank you for your professional comments. In the photoelectric test of this material, the equipment we use is a handheld dual-wavelength ultraviolet lamp (ENF-240C, Spectronics), the equipment power is 4w, when the test distance is 6 inches, it can provide a long-wave ultraviolet light intensity (UV-A) of 300μw/cm2, short-wave ultraviolet light intensity (UV-C) of 310μw/cm2. The degree of ultraviolet absorption shown in the ultraviolet absorption spectrum is a comprehensive embodiment of the part that ultraviolet light can penetrate the device and is sensed by the device. Due to the weak penetration ability of 254nm ultraviolet light, the light transmittance of the device is low, and the power of the device is too small, resulting in a high absorption rate of ultraviolet light at 254nm in Figure 5(a). We sincerely hope that the answers would meet your requirements. Thanks again for your constructive comments.

Comments on the Quality of English Language: English requires extensive polishing to enhance clarity and coherence.

Answer: Thank you for your professional comments. We feel sorry for our poor writings, however, we do invite a friend of us who is a native English speaker from the US to help polish our article. And we hope the revised manuscript could be acceptable for you. Thanks again for your comments.

Round 2

Reviewer 1 Report

A revised manuscript can be accepted for publication

Author Response

Thanks very much.

Reviewer 2 Report

The revised manuscript has indeed seen some improvements, but it is apparent that certain revisions and responses to the original review report remain perplexing and incongruous. Therefore, an additional revision is needed to rectify these issues and ensure that the manuscript achieves the desired level of clarity and cohesiveness.

1. In the response, the authors explained “complexation of photogenerated electron-hole pairs” as a process where photo-generated electron-hole pairs (i.e., excitons) recombine with another electron, which is impossible. I am guessing the authors meant to discuss the recombination (or decay) of holes and electrons?

2. I am glad the authors added a review of related works in the introduction; however, it could be revised to be more concise.

3. Regarding the band diagram of the TiO2/PANI heterostructure, it's not necessary to present how the built-in electric field is formed. The essential idea is to show how the electron-hole pairs are generated and separated at the interface and in which direction they move after separation.

4. Regarding the response to my original question about FTO, why would FTO increase the ultraviolet light absorption of the device? As shown in Figure 1, FTO is at the very bottom (underneath Ti), and the light is incident from the top. How could FTO at the very bottom contribute to the absorption of the device?

5. The response to "The absence of response at reverse bias" is incorrect. A photodiode normally works at reverse bias, as external bias enlarges the potential (built-in field) at the depletion region, allowing for easier separation of photo-generated electron-hole pairs. However, this is not seen in Figure 5b.

6. The equation for responsivity is incorrect. It should include only power in the denominator (or intensity times sensor area), not power times area.

7. In Figure 6a, where did the photocurrent come from in the TiO2 NTA-only sample at 0 bias? There's no electric field driving the transfer of photogenerated charge carriers.

Extensive editing of the English language is still needed to improve overall readability.

Author Response

List of Responses to Editor and Reviewers

Dear Editor and Reviewers,

Thank you very much for your constructive comments and suggestions concerning our article entitled “Preparation and photovoltaic performance of TiO2/PANI nanotube arrays composite UV photodetector” (Manuscript ID: polymers-2560727), which are all valuable and very helpful for revising and improving the quality of the manuscript. According to these comments and suggestions, we have made extensive modifications to our manuscript and supplemented extra data to make our results convincing. Revised portion are marked highlight with red colored words in the revised manuscript. The main corrections in the revised manuscript and the responses to your comments are as flowing:

Reviewer #2:

The revised manuscript has indeed seen some improvements, but it is apparent that certain revisions and responses to the original review report remain perplexing and incongruous. Therefore, an additional revision is needed to rectify these issues and ensure that the manuscript achieves the desired level of clarity and cohesiveness.

Answer: Thank you for your comments. We feel grateful for your professional review work on our article. We are sorry that our last reply did not fully address your doubts. As you are concerned, there are still some questions that need to be addressed. Based on your suggestions, we have revised the current manuscript again, hoping that the revised manuscript will achieve the clarity and coherence you expect. The detailed corrections are listed below. Thank you very much.

1.In the response, the authors explained “complexation of photogenerated electron-hole pairs” as a process where photo-generated electron-hole pairs (i.e., excitons) recombine with another electron, which is impossible. I am guessing the authors meant to discuss the recombination (or decay) of holes and electrons?

Answer: Thank you for your professional comments. We appreciate your highlighting the inadequacy of the pertinent representations in the manuscript. The discussion of the "photogenerated electron-hole pair" recombination has been updated, and it has been given a red font highlight in the manuscript. We sincerely hope that the answers would meet your requirements. Thanks again for your comments.

  1. I am glad the authors added a review of related works in the introduction; however, it could be revised to be more concise.

Answer: Thank you for your comments. Based on your suggestion, we have refined and simplified the introductory section, which we have revised and highlighted in the manuscript. Hope to meet your requirements. Thanks again for your professional comments.

Original:

In recent years, UV photodetectors have gradually become a hot spot in the field of scientific research [1,2]. UV photodetectors in the mid-ultraviolet and near-ultraviolet wavelength bands have widely used in major fields, such as military and defense [3], biomedicine [4,5], environmental monitoring [6], and light wave communication [7] due to their excellent performance in high sensitivity, wide linear range, and good stability. The use of ultraviolet rays in sunlight or flames for ozone holes and fire monitoring has gradually become the focus of research. Compared with traditional infrared and other types of photodetectors, UV photodetectors show remarkable ad-vantages, such as excellent photosensitivity and accuracy, strong concealment, high military value, and fast presentation of detection results [8-10]. Despite the numerous researches conducted on UV photodetectors, there are still many unsolved problems, such as the difficulty and high cost of preparing wide-band semiconductor arrays, the poor stability of the material itself, and the insufficient performance of the devices [11-13]. To meet the needs of various fields, high-performance, high-stability, high-reliability UV photodetectors, especially UV detectors that can work under ex-treme and harsh conditions, are extremely important, and the reliability and accuracy of UV photode-tectors have a crucial impact on practical applications.

At present, a variety of semiconductor materials with different bandwidths have been used as substrate materials for UV photodetectors due to their own advantages [14-20]. TiO2 has the advantages of good chemical stability, low toxicity, environmental friendliness, and it is one of the most widely used wide bandwidth semiconductors nowadays [21-23]. Among them, the rutile TiO2 has a bandwidth of about 3.0 eV, and the anatase TiO2 has a bandwidth of about 3.2 eV [24]. Because of the valence band energy difference between the two crystal phases, the heterojunction constructed by combining the two can reduce the forbidden bandwidth, resulting in a mixed crystal effect [25], which can promote the separation of photogenerated electrons and hole pairs to a certain extent. The above excellent optoelectronic properties have attracted much attention in ultraviolet photodetection [26-27]. Among them, TiO2 NTAs are highly ordered in structure, which can provide numerous charge transport channels, a larger specific surface area, and more interaction sites with UV light, providing great possibilities for the design and manufacture of higher-performance and high-er-efficiency UV photodetectors [28-32]. However, TiO2 as an n-type metal oxide semiconductor has a very high internal charge complex rate, and TiO2-based photodetectors require an external voltage to provide potential difference excitation for proper operation, which leads to large device size and severely limits their application in real life [33].

Polyaniline (PANI) is a common p-type conductive polymer with excellent electrical conductivity and electrochemical properties. Meanwhile, it has the advantages of high capacitance, high carrier mobility, high light absorption coefficient, ease of syn-thesis, and good environmental friendliness [34-37]. It is well known that the built-in electric field of p-n heterojunction can induce photoelectrons and holes to be transferred in opposite directions, which can achieve the purpose of inhibiting photogenerated carriers from compounding with each other [38,39]. Therefore, the p-n hetero-junction can be constructed by effectively compounding the n-type metal oxide semi-conductor TiO2 with the p-type conductive polymer PANI [40-42], and the potential difference automatically formed inside the heterojunction can provide the impetus for the separation of photogenerated electron-hole pairs and the formation of photocurrent, which can make the UV photodetectors have excellent UV sensitivity [43-45].

The microstructure of TiO2 nanotube arrays (NTAs) is neatly arranged nanotubes and fixed on the titanium substrate, which has a large specific surface area, good stability, and easy recyclability. However, due to the large band gap, response to ultraviolet light, and high electron-hole pair recombination rate of TiO2, it is necessary to improve the performance of TiO2 materials by modifying doping. At present, many researchers have cleverly compounded TiO2 NTAs with conductive polymer PANI to obtain countless materials with excellent performance. Wang et al. [47] prepared a TiO2 nanotube array photoelectrode (PANI/TiO2) wrapped by polyaniline (PANI) nanosheets by simple anodizing method and vacuum-assisted impregnation method. The results show that when the polyaniline nanosheets are tightly wrapped in TiO2 nanotubes, they have higher light response and charge separation efficiency than pure TiO2. Among them, the photocurrent density of PANI/TiO2-0.1 is 0.73 times that of TiO2. On PANI/TiO2-0.1, the degradation activity of PEC p-phenol was 27.7% higher than that of TiO2. It was proved that the synergistic effect of photocatalysis and electrocatalysis was generated on the polyaniline/TiO2 photoelectrode. Chen et al. [51] prepared a PANI/H-TiO2 nanotube array composite electrode by constant voltage deposition in electrochemical deposition. They first obtained the titanium dioxide nanotube array by secondary anodizing of the metal titanium sheet, then calcined and hydrogenated it in a hydrogen environment, and finally placed the hydrogenated sample in acetone solution containing sulfuric acid and aniline, and deposited polyaniline on the titanium dioxide nanotube array by constant voltage method to obtain a PANI/H-TiO2 nanotube array composite electrode. They measured the solution resistance and charge transfer resistance of 0.1554 and 2.723 V cm2, respectively, indicating that PANI/H-TiO2 nano-tubes have better electron transport ability and high charge transfer rate. Xie et al. [52] prepared a PANI-TiO2 nanotube array composite electrode using the potentiostatic potential polymerization method in electrochemistry. First, the anodized titanium di-oxide nanotube array is immersed in a hydrochloric acid solution containing aniline monomer, stirred, and fully infiltrated, and then polyaniline is deposited on the nano-tubes by potentiostat method (voltage 0.9 V) in the same solution to obtain a composite electrode. They observed that TiO2 nanotubes were wrapped with disordered polyaniline nanowire arrays, with unique microstructure, high specific surface area, fast ion diffusion path, and long-term cycle stability, and the composite showed excellent electrochemical performance. It can be seen that after the recombination of TiO2 nanotube array and PANI, the improvement of specific surface area, stability, and charge separation efficiency of the material is extraordinary, and it has great development prospects in the field of photoelectric detection materials in the future.

Based on the work of other researchers in this field, this work analyzes the current status of photodetectors and proposes to use a simpler two-step electrochemical method to composite TiO2 nanotube arrays with P-type conjugated conductive polymer PANI, to further expand the spectral response range of the material and increase the device's absorption of ultraviolet light, At the same time, a good p-n heterojunction interface can be formed inside the device, which is conducive to accelerating the separation of photogenerated electron hole pairs, suppressing carrier recombination, effectively improving quantum efficiency, and further improving the response speed, sensitivity, and stability of ultraviolet photodetectors.

Revised:

UV photodetectors have increasingly emerged as a popular topic in science research in recent years [1,2]. Due to their excellent performance in high sensitivity, wide linear range, and good stability, UV photodetectors in the mid-ultraviolet and near-ultraviolet wavelength bands have been widely used in important fields, including military and defense [3], biomedicine [4,5], environmental monitoring [6], and light wave communication [7]. The focus of study has steadily shifted to the utilization of UV radiation from fires or sunlight for ozone hole and fire monitoring. UV photodetectors have notable advantages over conventional infrared and other types of photodetectors, including outstanding photosensitivity and accuracy, good concealment, high military value, and quick presenting of detection data [8–10]. The preparation of wide-band semiconductor arrays, which is complex and expensive, the material's poor stability, and the devices' inadequate performance are just a few of the issues that remain unresolved despite the extensive research on UV photodetectors [11–13]. High-performance, high-stability, high-reliability UV photodetectors, particularly UV detectors that can operate in extreme and harsh conditions, are extremely important to meet the needs of various fields, and the accuracy and reliability of UV photodetectors have a significant influence on practical applications.

Due to their benefits, several semiconductor materials with various bandwidths are currently being employed as substrate materials for UV photodetectors [14–20]. TiO2 is one of today's most popular wide-bandwidth semiconductors due to its benefits of strong chemical stability, low toxicity, and environmental friendliness [21–23]. Among these, the rutile TiO2 and the anatase TiO2 have bandwidths of around 3.0 and 3.2 eV, respectively [24]. The heterojunction formed by combining the two crystal phases can reduce the forbidden bandwidth and produce a mixed crystal effect [25], which can somewhat facilitate the separation of photogenerated electron and hole pairs. This is because the two crystal phases have different valence band energies. In ultraviolet photodetection, the aforementioned good optoelectronic features have received a lot of attention [26–27]. Among them, TiO2 NTAs have a highly ordered structure that can offer numerous charge transport channels, a greater specific surface area, and more UV light interaction sites, opening up a wide range of possibilities for the development of UV photodetectors with higher performance and efficiency [28–32]. TiO2, on the other hand, has a very high internal charge complex rate as an n-type metal oxide semiconductor, and TiO2-based photodetectors need an external voltage to provide potential difference excitation for proper operation, which results in large device size and severely restricts their application in real life [33].

P-type conductive polymers like polyaniline (PANI) have high electrical conductivity and electrochemical characteristics. It absorbs light well, moves carriers quickly, has a high capacitance, is easy to fabricate, and is environmentally friendly [34–37]. Photogenerated carriers can be prevented from combining in p-n heterojunctions because their internal electric fields cause photoelectrons and holes to flow in different directions [38,39]. Mixing the n-type metal oxide semiconductor TiO2 with the p-type conductive polymer PANI creates the p-n hetero-junction [40–42]. The heterojunction's automatically produced potential difference can push photogenerated electron-hole pairs apart and make photocurrent, giving UV photodetectors [43–45] high UV sensitivity.

The titanium substrate, which has a large specific surface area, stability, and recycling ease, is fixed with neatly organized TiO2 nanotube arrays (NTAs). However, TiO2's large band gap, UV light responsiveness, and high electron-hole pair recombination rate require doping changes to improve performance. Numerous researchers have coupled TiO2 NTAs with the conductive polymer PANI to develop many high-performance compounds. Wang et al. [47] produced a TiO2 nanotube array photoelectrode (PANI/TiO2) wrapped in polyaniline (PANI) nanosheets using anodizing and vacuum-assisted impregnation. Polyaniline nanosheets that are securely wrapped in TiO2 nanotubes exhibit superior light responsiveness and charge separation efficiency than pure TiO2. PANI/TiO2-0.1 has 0.73 times the photocurrent density of TiO2. PEC p-phenol degrades PANI/TiO2-0.1 27.7% better than TiO2. It was found that the polyaniline/TiO2 photoelectrode synergized photocatalysis and electrocatalysis. Chen et al. [51] produced a PANI/H-TiO2 nanotube array composite electrode utilizing constant voltage electrochemical deposition. Before making a PANI/H-TiO2 nanotube array composite electrode, the researchers secondary anodized titanium sheet. They then calcined and hydrogenated the hydrogenated sample in hydrogen, put it in an acetone solution containing sulfuric acid and aniline, and deposited polyaniline on the titanium dioxide nanotube array using continuous voltage. PANI/H-TiO2 nanotubes exhibit excellent electron transport and charge transfer, as shown by their solution resistance of 0.1554 and charge transfer resistance of 2.723 V cm2. Xie et al. [52] used electrochemical potentiostatic potential polymerization to make a PANI-TiO2 nanotube array composite electrode. The anodized titanium dioxide nanotube array is immersed in a hydrochloric acid solution with aniline monomer, agitated until completely dissolved, and then polyaniline is added using the potentiostat method (0.9 V) in the same solution to generate a composite electrode. They found arrays of disordered polyaniline nanowires around TiO2 nanotubes, which possessed a unique microstructure, high specific surface area, fast ion transport channel, and long-term cycle stability. Excellent electrochemical performance was shown by the composite. After recombining the TiO2 nanotube array and PANI, the material's specific surface area, stability, and charge separation efficiency improved. Future photoelectric detection materials could benefit from this substance.

This study examines photodetectors and suggests combining TiO2 nanotube arrays with P-type conjugated conductive polymer PANI using a two-step electrochemical method to increase the material's spectral response range and ultraviolet light absorption. This analysis relies on previous research. A good p-n heterojunction interface inside the device improves ultraviolet photodetectors' response time, sensitivity, and stability, accelerates photogenerated electron hole pair separation, suppresses carrier recombination, and boosts quantum efficiency.

  1. Regarding the band diagram of the TiO2/PANI heterostructure, it's not necessary to present how the built-in electric field is formed. The essential idea is to show how the electron-hole pairs are generated and separated at the interface and in which direction they move after separation.

Answer: Thank you for your professional comments. Based on your suggestions, we further introduce in detail the mechanism of separation of photogenerated electron-hole pairs by TiO2 NTAs/PANI composites in figure 8, as well as the direction of movement of electrons and holes after separation. We sincerely hope that the answers would meet your requirements. Thanks again for your professional comments.

Original:

Figure 8. The contact potential differences of (a) PANI and (b) TiO2.

Revised:

Figure 8. The contact potential differences of (a) PANI and (b) TiO2, (c) Schematic diagram of interface electron transfer

  1. Regarding the response to my original question about FTO, why would FTO increase the ultraviolet light absorption of the device? As shown in Figure 1, FTO is at the very bottom (underneath Ti), and the light is incident from the top. How could FTO at the very bottom contribute to the absorption of the device?

Answer: Thank you for your professional comments. The most important function of FTO placed at the bottom is to facilitate the fixation and support of the UV photodetector device and at the same time act as the electrode or current collector part of the device. The problem that FTO helps absorb light is the result of comparison with other opaque materials. Compared with other opaque substrate materials, the choice of FTO with good light transmittance can promote light absorption of the device to a certain extent. However, this part of the effect is relatively weak, and most of the light source still enters from the top of the device. We hope that our explanation can be accepted by you and thank you again for your very professional and meticulous comments.

  1. The response to "The absence of response at reverse bias" is incorrect. A photodiode normally works at reverse bias, as external bias enlarges the potential (built-in field) at the depletion region, allowing for easier separation of photo-generated electron-hole pairs. However, this is not seen in Figure 5b.

Answer: Thank you for your professional comments. Thank you for pointing out the typo in the manuscript. We agree with you that photodiodes typically operate with reverse bias because this amplifies the potential in the built-in field of the depletion region, which facilitates the separation of the electron-hole pairs generated by the light. Due to the low power of the UV light source utilized in the test, which is weaker than the forward bias, the photocurrent in Figure 5b is reduced when reverse-biased. Based on the initial design, the local magnification of the reverse bias time current has now been introduced, making it easier to understand and see how the device performs photoelectrically. It is not difficult to see that the photocurrent grows with the wavelength of the irradiated ultraviolet light under the same reverse bias settings and has good sensitivity to ultraviolet light. Its current is larger under ultraviolet light than in a dark environment. We sincerely hope that the answers would meet your requirements. Thanks again for your professional comments.

Original:

Figure 5. UV–vis light absorption spectra (a) and I–V curves (b) of TiO2 NTAs and TiO2 NTAs/PANI nanocomposite.

Revised:

Figure 5. UV–vis light absorption spectra (a) and I–V curves (b) of TiO2 NTAs and TiO2 NTAs/PANI nanocomposite.

  1. The equation for responsivity is incorrect. It should include only power in the denominator (or intensity times sensor area), not power times area.

Answer: Thank you for your professional comments. Thanks for pointing out the error regarding the responsiveness equation. We have modified the formulation of the responsiveness equation in the manuscript. It would also be added to you that the active light area tested by this working device is 1cm2, so it does not affect the light responsivity calculation results. Thanks again for your professional comments.

Original:

R = ∆I/PS

ΔI is the difference between the photocurrent and the dark current, P is the optical power, and S is the effective irradiation area.

Revised:

R = ∆I/P

ΔI is the difference between the photocurrent and the dark current, P is the optical power.

  1. In Figure 6a, where did the photocurrent come from in the TiO2 NTA-only sample at 0 bias? There's no electric field driving the transfer of photogenerated charge carriers.

Answer: Thank you for your comments. TiO2 is a common semiconductor that can achieve photoelectric conversion, when light with a wavelength of 387 nm or less hits the electrode surface of TiO2 nanomaterials, electrons (e-) on the valence band are excited to transition to the conduction band, leaving holes (h+) in the valence band. The reason why the external bias voltage is zero and the current is present is that the electrons in the TiO2 NTA sample illuminated by 365nm ultraviolet light absorb the energy of photons. This energy is equal to Planck's constant multiplied by the frequency of light, and if it exceeds the escape work of the electron, it will separate the electron from the metal, and the excess part is the kinetic energy of the electron. This part of the electrons has kinetic energy, so it generates an electric current. However, because the photogenerated electrons and holes of TiO2 are easily reorganized, the photocurrent generated is weak, and its I-t curve will take on a sharper shape, as shown in Figure 6a. We sincerely hope that the answers would meet your requirements. Thanks again for your valuable comments.

Figure 6a. I-t curves of different devices at 0V bias under 365nmUV light irradiation

Comments on the Quality of English Language: Extensive editing of the English language is still needed to improve overall readability.

Answer: Thank you for your professional comments. We apologize for the lack of technical English in the manuscript. In this revision, we have further polished the entire text more comprehensively to ensure the consistency and accuracy of the sentences in the manuscript. We hope the revised manuscript could be acceptable for you. Thanks again for your comments.

Round 3

Reviewer 2 Report

Upon reviewing the revised manuscript, I am glad that the authors have made significant improvements to the quality and comprehensibility of their research. However, I must express my reservations regarding their response to my last comment, regarding the generation of photocurrent in TiO2 without an external bias voltage.

I agree that electrons can be excited into the conduction band in TiO2 after absorbing UV photons. However, without an external bias voltage (electric field), the formation of a photocurrent seems impossible, as it violates Ohm's Law  j=σE. The unaddressed issue of explaining the observed photocurrent at 0 bias in the TiO2-only sample raises concerns about the fundamental validity of the research. In light of this, I would like to extend another opportunity to the authors for revision. I hope the authors can provide a convincing and scientifically sound explanation for the photocurrent observed at 0 bias in the TiO2-only sample, as this is essential for reinforcing the manuscript's scientific credibility.

Minor editing of English language is required.

Author Response

List of Responses to Editor and Reviewers

Dear Editor and Reviewers,

Thank you very much for your constructive comments and suggestions concerning our article entitled “Preparation and photovoltaic performance of TiO2/PANI nanotube arrays composite UV photodetector” (Manuscript ID: polymers-2560727), which are all valuable and very helpful for revising and improving the quality of the manuscript. According to these comments and suggestions, we have made extensive modifications to our manuscript to make our results convincing. Revised portion are marked highlight with red colored words in the revised manuscript. The main corrections in the revised manuscript and the responses to your comments are as flowing:

Reviewer #2:

Upon reviewing the revised manuscript, I am glad that the authors have made significant improvements to the quality and comprehensibility of their research. However, I must express my reservations regarding their response to my last comment, regarding the generation of photocurrent in TiO2 without an external bias voltage.

I agree that electrons can be excited into the conduction band in TiO2 after absorbing UV photons. However, without an external bias voltage (electric field), the formation of a photocurrent seems impossible, as it violates Ohm's Law j=σE. The unaddressed issue of explaining the observed photocurrent at 0 bias in the TiO2-only sample raises concerns about the fundamental validity of the research. In light of this, I would like to extend another opportunity to the authors for revision. I hope the authors can provide a convincing and scientifically sound explanation for the photocurrent observed at 0 bias in the TiO2-only sample, as this is essential for reinforcing the manuscript's scientific credibility.

Answer: We appreciate you taking the time to share your thoughts and the expert review work you did on our article. We apologize if our previous response did not adequately allay your concerns. First and foremost, we would like to express our gratitude to the reviewers for their thorough and insightful comments, which have helped us greatly in enhancing the manuscript's scientific merit. We also conducted a thorough analysis of the research findings and looked into pertinent scholarly literature based on the reviewers' insightful comments. We hypothesize that the heterojunction between pure TiO2 and the current collector is the reason that pure TiO2 materials also produce photocurrent. In our test, the TiO2, Ti substrate, and FTO contacts make up the majority of the heterojunctions that may interact with photogenerated electrons. A faint photocurrent can form because FTO is a fluorine-doped SnO2 (tin dioxide) film that forms a very weak heterojunction at the point where it comes into contact with the TiO2 nanotube layer. Additionally, between the TiO2 nanotube layer and the substrate Ti sheet layer, there will be a metal/semiconductor metal oxide heterojunction that has the potential to create an integrated electric field. The weak photocurrent is observed at 0V bias in pure metal-oxide-semiconductor devices, which is consistent with many other similar device architectures.

For the TiO2 NW array device, for instance, Xihong Zu [1] and colleagues reported that the photocurrent grew to around 1.5nA at a bias of 0V by irradiating with 365nm UV light, then progressively decreased when the UV light was turned off (curve a). About 18 seconds passed between the times of the increase and decay, respectively. The photosensitivity and photocurrent of the device based on TiO2 NWs/PANI/TiO2 NWs heterostructured arrays were both significantly enhanced when compared to a bare TiO2 NW array. To demonstrate that pure TiO2 NTAs can produce photocurrent, Qing Kang et al. [2] measured the linear scanning voltammetry (LSV, 10 mV s 1) of raw and reduced TiO2 NTAs in 1 M NaOH solution under ultraviolet-visible (0.6 W cm2) irradiation as well as the change of photocurrent density of reduced TiO2 NTAs at 1.23 VRHE with NaBH4 processing time. At the same time, P Chinnamuthu et al. [3] thoroughly described the conduction mechanism of unipolar Schottky junctions and hypothesized that the large Auger effect would be to blame for the low current conversion at shorter wavelengths.

Furthermore, based on the SEM findings, we hypothesize that the TiO2 nanotube layer and the FTO layer have stratum areas that are significantly less than those of the TiO2 NTAs/PANI device, resulting in a reduced built-in electric field. The built-in electric field of pure TiO2 is much weaker than the built-in electric field of TiO2 NTAs/PANI devices, so the photocurrent is smaller when testing the photocurrent of pure TiO2 devices. In conclusion, even pure TiO2 devices have an internal electric field when testing photocurrent, which can generate photocurrent. Once again, we appreciate your thorough review, and we hope that our justification satisfies you.

Comments on the Quality of English Language: Minor editing of English language is required.

Answer: Thank you for your professional comments. In this revision, we have further carefully edited the linguistic expression of the manuscript to ensure the consistency and accuracy of the sentences in the manuscript. And we hope the revised manuscript could be acceptable for you. Thanks again for your comments.

References

[1] Zu, X.; Wang, H.; Yi, G.; Zhang, Z.; Jiang, X.; Gong, J.; Luo, H. Self-Powered UV Photodetector Based on Heterostructured TiO2 Nanowire Arrays and Polyaniline Nanoflower Arrays. Synthetic Metals 2015, 200, 58–65, http://doi.org/10.1016/j.synthmet.2014.12.030.

[2] Kang, Q.; Cao, J.; Zhang, Y.; Liu, L.; Xu, H.; Ye, J. Reduced TiO2 Nanotube Arrays for Photoelectrochemical Water Splitting. Journal of Materials Chemistry A 2013, 1, 5766, http://doi.org/10.1039/c3ta10689f.

[3] Chinnamuthu, P.; Dhar, J.C.; Mondal, A.; Bhattacharyya, A.; Singh, N.K. Ultraviolet Detection Using TiO2 Nanowire Array with Ag Schottky Contact. Journal of Physics D: Applied Physics 2012, 45(13),135102, http://doi.org/10.1088/0022-3727/45/13/135102.

Round 4

Reviewer 2 Report

The response from the authors is satisfactory. I recommend accepting the manuscript for publication with the condition that the text from the response be summarized and inserted into the main text. Once this is complete, the manuscript should be ready for final acceptance for publication.

Author Response

List of Responses to Editor and Reviewers

Dear Editor and Reviewers,

Thank you very much for your constructive comments and suggestions concerning our article entitled “Preparation and photovoltaic performance of TiO2/PANI nanotube arrays composite UV photodetector” (Manuscript ID: polymers-2560727), which are all valuable and very helpful for revising and improving the quality of the manuscript. According to your comments and suggestions, we have made extensive modifications to our manuscript to make our results convincing. Revised portion are marked highlight with red colored words in the revised manuscript. The main corrections in the revised manuscript and the responses to your comments are as flowing:

Reviewer #2:

The response from the authors is satisfactory. I recommend accepting the manuscript for publication with the condition that the text from the response be summarized and inserted into the main text. Once this is complete, the manuscript should be ready for final acceptance for publication.

Answer: Thank you very much for your valuable review feedback. We are very excited about your willingness to receive this manuscript. Thank you for your help and support in this work during this period! In this revision, we have added explanations for the photocurrent generated by pure TiO2 in the manuscript, as well as relevant references. Thanks again for your professional comments.

Revised:

In Figure6(c), we hypothesize that the heterojunction between pure TiO2 and the current collector is the reason that pure TiO2 materials also produce photocurrent. In our test, the TiO2, Ti substrate, and FTO contacts make up the majority of the heterojunctions that may interact with photogenerated electrons. A faint photocurrent can form because FTO is a fluorine-doped SnO2 (tin dioxide) film that forms a very weak heterojunction at the point where it comes into contact with the TiO2 nanotube layer. Additionally, between the TiO2 nanotube layer and the substrate Ti sheet layer, there will be a metal/semiconductor metal oxide heterojunction that has the potential to create an integrated electric field[57-59]. The weak photocurrent is observed at 0V bias in pure metal-oxide-semiconductor devices, which is consistent with many other similar device architectures. Furthermore, based on the SEM findings, we hypothesize that the TiO2 nanotube layer and the FTO layer have stratum areas that are significantly less than those of the TiO2 NTAs/PANI device, resulting in a reduced built-in electric field. The built-in electric field of pure TiO2 is much weaker than the built-in electric field of TiO2 NTAs/PANI devices, so the photocurrent is smaller when testing the photocurrent of pure TiO2 devices. In conclusion, even pure TiO2 devices have an internal electric field when testing photocurrent, which can generate photocurrent.

References

[57] Zu, X.; Wang, H.; Yi, G.; Zhang, Z.; Jiang, X.; Gong, J.; Luo, H. Self-Powered UV Photodetector Based on Heterostructured TiO2 Nanowire Arrays and Polyaniline Nanoflower Arrays. Synthetic Metals 2015, 200, 58–65, http://doi.org/10.1016/j.synthmet.2014.12.030.

[58] Kang, Q.; Cao, J.; Zhang, Y.; Liu, L.; Xu, H.; Ye, J. Reduced TiO2 Nanotube Arrays for Photoelectrochemical Water Splitting. Journal of Materials Chemistry A 2013, 1, 5766, http://doi.org/10.1039/c3ta10689f.

[59] Chinnamuthu, P.; Dhar, J.C.; Mondal, A.; Bhattacharyya, A.; Singh, N.K. Ultraviolet Detection Using TiO2 Nanowire Array with Ag Schottky Contact. Journal of Physics D: Applied Physics 2012, 45(13),135102, http://doi.org/10.1088/0022-3727/45/13/135102.